# The Pandemic Journaling Project: A new dataset of first-person accounts of the COVID-19 pandemic

Sarah S. Willen[1,2], Katherine A. Mason[3,4], Heather M. Wurtz,[5] Sebastian Karcher[6,7]*

1 Department of Anthropology, University of Connecticut, Storrs, Connecticut, United States of America, 2 Research Program on Global Health and Human Rights, Gladstein Family Human Rights Institute, University of Connecticut, Storrs, Connecticut, United States of America, 3 Department of Anthropology, Brown University, Providence, Rhode Island, United States of America, 4 Population Studies and Training Center, Brown University, Providence, Rhode Island, United States of America, 5 Department of Population and Public Health Sciences, University of Southern California, Keck School of Medicine, Los Angeles, California, United States of America, 6 Department of Political Science, Syracuse University, New York, United States of America, 7 Qualitative Data Repository, Syracuse University, New York, United States of America

* skarcher@syr.edu

## Abstract

### Background

The rapid global spread of the COVID-19 pandemic affected different regions, communities, and individuals in vastly different ways that interdisciplinary social scientists are well-positioned to document and investigate. This paper describes an innovative mixed-methods dataset generated by a research study that was designed to chronicle and preserve evidence of the pandemic's divergent effects: the Pandemic Journaling Project (PJP). The dataset was generated by leveraging digital technology to invite ordinary people around the world to document the impact of the COVID-19 pandemic on their everyday lives over a two-year period (May 2020-May 2022) using text, images, and audio.

### Methods and findings

PJP's weekly, online, bilingual (English/Spanish) journaling platform was open to anyone with access to a smartphone or computer, including teens aged 15–17 with permission of a parent or guardian. Participants first completed a baseline quantitative survey, after which they were invited to create two journal entries in response to suggested narrative prompts. In each subsequent week, participants received weekly invitations to contribute via their choice of email or SMS (text message). Each invitation included a link to that week's journaling prompts and accompanying survey questions.

**Data availability statement:** Data described in this paper are available at the Qualitative Data Repository as Willen, Sarah S.; Mason, Katherine A.. 2024. "Data for: The Pandemic Journaling Project, Phase One (PJP-1)". Qualitative Data Repository. https://doi.org/10.5064/F6PXS9ZK Since data contains identifiable human participants, and as required by the UConn IRB, access to full data is by application. Access to the data can be requested via the "Request Access" button on the dataset's landing page and follows the policy described in the terms of access document (https://doi.org/10.5064/F6PXS9ZK/7UYI4F). An unrestricted dataset that includes the summary data and analysis code that was used to produce the figures in the paper is available on the Harvard Dataverse at https://doi.org/10.7910/DVN/QTQ3V7.

**Funding:** The authors thank the multiple sponsors of the Pandemic Journaling Project at the University of Connecticut (SSW) and Brown University (KAM), including the Office of the Vice President for Research at UConn (https://ovpr.uconn.edu/) and Brown (https://division-research.brown.edu/), as well as the Human Rights Institute (https://human-rights.uconn.edu/), Humanities Institute (https://humanities.uconn.edu/), Institute for Collaboration on Health, Intervention, and Policy (InCHIP: https://chip.uconn.edu/), and Office of Global Affairs (https://global.uconn.edu/) at the University of Connecticut, and also the Population Studies and Training Center (supported by NIH center grant P2C HD041020; https://pstc.brown.edu/) and Department of Anthropology (https://anthropology.brown.edu/) at Brown University. A full list of sponsors is available at https://doi.org/10.5064/F6PXS9ZK/F2LRNE. No sponsors or funders played any role in the study design, data collection and analysis, decision to publish, or preparation of the manuscript. SK's work was supported by a grant from the National Science Foundation (2116935).

**Competing interests:** The authors have declared that no competing interests exist.

Participants could join the project at any point between May 2020 and May 2022. PJP employed a cohort design. Regardless of when they joined, all received the narrative prompts and accompanying survey questions in the same order. Participants could stop participating at any point, and they could later restart if they wished. The project welcomed any interested participant and sought to capture as broad a range of perspectives as possible, while also taking measures to include voices that might not otherwise be preserved in the historical record.

The project launched in May 2020. In the two years it operated as a weekly journaling platform, PJP generated nearly 27,000 individual journal entries from over 1,800 people in 55 countries around the world. Data from PJP's first phase (PJP-1) are now accessible at the Qualitative Data Repository (QDR) at Syracuse University.

## Conclusions

The first phase of the Pandemic Journaling Project has produced an innovative multi-media dataset that can support studies of how the COVID-19 pandemic has affected a wide range of communities across a wide range of outcomes including mental health, reproductive health, vaccine hesitancy, and trust in health professionals, among others. The dataset is available to researchers who follow established data protection protocols and procedures. These data protection measures will be in place for 25 years, through 2049, after which all PJP-1 data will become a fully accessible public archive via QDR.

## Introduction

When the COVID-19 pandemic began spreading rapidly around the globe, experts struggled to understand the epidemiology of the virus [1,2]; to produce narratives that anxious publics could understand [3,4]; and to generate practical guidelines that could interrupt viral transmission and minimize morbidity and mortality [5–7]. As this new virus jumped across borders, countries, and continents, it quickly became clear that different regions, communities, and individuals would be affected in vastly different ways [8–10]. In that confusing moment, we found ourselves asking: How could these divergent effects be chronicled and preserved? This paper describes a mixed-methods dataset generated by a research study designed to address precisely this question: the Pandemic Journaling Project (PJP).

Created by medical anthropologists with support from an interdisciplinary team of collaborators, PJP was designed to give ordinary people around the world a chance to chronicle the ongoing impact of the COVID-19 pandemic on their lives, in real time, using a smartphone or other electronic device. The platform was fully bilingual in English and Spanish, but participants could create their entries in any language. The project launched in May 2020, just weeks after the World Health Organization declared COVID-19 a pandemic. In the two years PJP operated as a weekly journaling platform, it generated a comprehensive trove of nearly 27,000 individual journal entries from over 1,800 people in 55 countries around the world.

Data from PJP's first phase (PJP-1) are now accessible at the Qualitative Data Repository (QDR) at Syracuse University. [11] For a period of 25 years, through 2049, PJP-1 data will be accessible via QDR to interested researchers who obtain appropriate permissions, follow QDR's standard data protection protocols, and follow the procedures and policies outlined in PJP's Terms of Access Agreement and in accordance with their own institutional IRBs. From 2049 onward, all PJP-1 data will become a fully accessible public archive via QDR.

A number of features distinguish PJP from other COVID-19 archival projects [see, e.g., 12–16]. First and foremost is its dual mission. PJP was designed to accomplish a key research goal while also providing meaningful benefit for research participants. As a research endeavor, PJP's primary objective was to "pre-design an archive" of COVID-19 narratives and experiences that would provide health researchers, and scholars in other fields, fine-grained insight into the impact of the COVID-19 pandemic on individual, family, and communal lives. For participants, it provided a chance to take a few minutes each week to reflect on how the pandemic was affecting their lives. Each new entry became part of a weekly journal that participants could download securely at any point and keep for themselves, while also knowing that their journals would become part of the growing collaborative archive we were building together.

A second distinguishing feature of PJP is its multimedia nature. Journal entries could be submitted in text format, as images, or as audio journal entries. Not only did the project's broad definition of journaling yield a multimedia trove of data, but it also encouraged participation from participants who were interested in chronicling their experience but preferred to express themselves in modalities other than writing.

A third key feature of PJP is its mixed-methods design. To supplement their qualitative journal entries, participants completed a comprehensive baseline survey upon enrollment that collected self-reported data on a broad range of topics including demographics; health status (on physical and mental health); COVID-19 exposure and mitigation; health insurance; political leanings; media consumption practices; and degree of trust in government and institutions. Some quantitative measures (e.g., items on health, COVID-19 exposure and mitigation, and trust in government) were then repeated in subsequent weeks, yielding insight into change over time.

Other key features of PJP include its longitudinal approach — and its global reach. Unlike COVID-19 projects that gathered data in a single location at a specific moment in time, [e.g., 12,13] PJP enabled participants to participate weekly for an extended period of weeks, months, or even years — and it drew participants from 55 countries. Finally, PJP is distinguished by equity-oriented and human-rights based commitments that influenced the data collection process and, as a result, the dataset itself. [17] The PJP interface was designed to be as accessible and inclusive as possible, and our recruitment efforts were crafted to encourage participation from minoritized communities whose perspectives might otherwise be neglected or overlooked. Participants were able to make decisions about what kind of knowledge was collected, and on what terms. In addition, they were able to access and keep the data they contributed by downloading their journals.

Below, we introduce this mixed-methods dataset, beginning with a discussion of the rationale for creating PJP and its overall goals and objectives. A detailed description of the research methods and materials included in this dataset follows. Here we pay particular attention to the team's unconventional decision, as anthropologists, to deposit our data in a publicly accessible data repository. We also highlight the formative role of the collaboration between PJP and QDR, which began during our initial study design and continued through the data collection process and into the construction and composition of the dataset. We then discuss the dataset itself, including the data gathered and the value of collecting multimedia materials. The subsequent section introduces some of the analyses already conducted using PJP-1 data, which explore the impact of COVID-19 on a range of specific groups and topics. We conclude with a discussion of the broader significance of the PJP-1 dataset and future analytic possibilities it opens up for health researchers in the near- and longer-term.

## "Pre-designing an archive": Interdisciplinary rationale

PJP's project of "pre-designing an archive" of COVID-19 narratives and experiences has three core dimensions. First, it enabled individual participants to create a personal archive that was theirs to keep and use as they wished. Second, it

generated a freely accessible, curated public archive — our "Featured Entries" page. [18] Finally, and most importantly from a scientific standpoint, it generated a collection of nearly 27,000 COVID-19 narratives and experiences that other researchers can access and analyze, now and well into the future.

Achieving this aim involved a multi-stage process that began with the project's inception in Spring 2020, against the backdrop of an unfolding global crisis. At that early point, the team quickly reached a number of key decisions, including decisions about (1) the analytic goals of the project; (2) its disciplinary and interdisciplinary commitments; (3) the need to balance research goals with a commitment to providing participants a meaningful opportunity to chronicle their experiences; (4) ethical considerations (for instance, protecting participants' confidentiality while ensuring they could securely access their own journals at any point); (5) data security; and (6) data curation and accessibility, both near-term and long-term. All of these decisions were made during a period of major disruption to the everyday operations of our respective institutions, institutional review boards, etc., and to the professional and personal lives of all involved research partners and potential participants. In conceptualizing the project and working through these decisions, we knew that the sooner the project launched, the more robust and useful the findings ultimately would be from a research standpoint. The preparation stage, from initial conceptualization through seed fundraising, IRB approval, and launch of the journaling platform in late May 2020, took nine weeks.

As an archival endeavor, PJP is grounded in a commitment to democratizing the process of knowledge production, and to the human right to access and participate in science. [17] According to United Nations Special Rapporteur on the Right to Science, Alexandra Xanthaki, science is a common good, and "the right to participate in science [is] an element of the right to participate in cultural life." [19] From this standpoint, the practice of science must be "based on the principles of the universality and indivisibility of rights, non-discrimination, equality, participation and respect for cultural diversity, including scientific diversity," and it should include "the democratization of science and its production." [ibid] Put differently, scientific endeavors must be designed to comport with human rights frameworks and to hold benefit for humanity writ large.

PJP pursues these goals using three main strategies: *grassroots collaborative ethnography, archival activism, and anticipatory archiving*. First, "grassroots collaborative ethnography" reflects our goal of empowering people to see their own stories as worth telling, and sharing. PJP has not only provided the means to do so in concrete ways, but we also have sought out opportunities to co-produce knowledge in collaboration with our interlocutors. [20,21] A second strategy, "archival activism," is rooted in our "strategic commitment to facilitating equitable history-telling in the future through inclusive collection and preservation practices in the present." [20,22] PJP's motto, posted on the project website, [23] points to this core commitment: "*Usually, history is written only by the powerful. When the history of COVID-19 is written, let's make sure that doesn't happen*." Finally, our approach to archival activism is rooted in a commitment to "anticipatory archiving," which has four dimensions:

(1) a recognition that the "now" has historic significance, (2) a plan to systematically document current events, (3) forward-looking reflection on how people might use the data in the future, and (4) a commitment to letting that future-oriented thinking shape collection efforts in the present. [24]

While the theoretical elaboration of these strategies emerged over time, the underlying commitments that define them were part of PJP's design from the outset and have thus informed the shape of the dataset. These decisions also mean that all data in PJP-1 are self-reported and should be analyzed using best practices for such self-reported health data [e.g., 25].

Since the project's launch, research participants have participated in the process of scientific knowledge production in various ways, but one dimension stands out in particular: data that participants generated has always belonged to them as much as it has belonged to PJP. Throughout the PJP-1 data collection process, participants could log in securely at any point to access, view, and download their journal entries via a user-friendly, secure interface. Beyond the data

creation and collection process, participants have been given as much control as possible over how, where, and when their contributions are used outside of the formal data creation and analysis process. For example, the public-facing "Featured Entries" page displays only journal entries that contributors have granted explicit permission to share in that space. Importantly, the Featured Entries page was designed to be different from social media in that visitors to the page can view (or, in the case of audio entries, hear) contributions, but the page offers no mechanism for direct interaction (i.e., "liking," commenting, or otherwise responding either publicly or privately, through direct messaging). By excluding interactive features that tend to be standard on social media sites, we created a space in which visitors could share personal experiences, and encounter those of others, without risk of judgment.

The team has also worked directly with contributors in creating a multimedia exhibition of PJP photos and audio clips that traveled to five cities in four countries in 2022−23 (*Picturing the Pandemic: Images from the Pandemic Journaling Project*; see https://picturingthepandemic.org/). The team cataloged and reviewed all audio and visual contributions, then selected a subset of materials for potential display. Materials included in the exhibition were only displayed after obtaining explicit permission from participants and asking participants for additional information about themselves or their contribution they might want to include on display. News of the exhibitions was shared with all PJP participants, and many who lived nearby attended a local launch or visited at a later opportunity.

## Methods and materials

Below we describe the methods involved in generating the PJP-1 dataset, including key decisions taken at the outset, followed by an overview of the materials deposited at QDR. PJP-1 was approved as a research study by the UConn-Storrs Institutional Review Board (IRB) at the University of Connecticut (Protocol #: HR20−0065). Participants provided written consent to participate. The consent forms in English and Spanish, which are published as part of the dataset, specify that consent to participate in PJP-1 included consent for materials contributed to be shared with other researchers and, in 25 years, as an ungated, publicly accessible archive.

## Methods

### Recruitment and sampling

PJP's bilingual platform was open to anyone with access to a smartphone or computer, including teens ages 15–17 with permission of a parent or guardian. Participants were introduced to the project in a variety of ways, among them the PJP website, professional networks, PJP's social media accounts (on Facebook, Instagram, and Twitter), faculty invitations to engage PJP in classroom settings, and media coverage of the project. Our Academic and Student Advisory Boards, who represent a wide range of racial/ethnic, socioeconomic, generational, and disciplinary backgrounds, also played key roles in disseminating news of PJP. Other key partners in dissemination included community organizations, high school administrators, and guidance counselors in our local areas, and beyond. We also gave community presentations and guest lectures about PJP in a wide range of settings, ranging from academic medical centers and university classrooms to urban charter schools and one religious congregation. We also supported several colleagues who were eager to introduce PJP in their own community-based research, teaching, and youth projects, including colleagues working with high school students in Black communities in South Africa, Black university students in Brazil, and urban youth in central Mexico. One year after the project's launch, in May 2021, we partnered with the Commission on Women, Children, Seniors, Equity, & Opportunity of the Connecticut State Legislature in organizing a virtual forum on, "Journaling During COVID Times." At the virtual forum and in other public venues, members of some of our target groups (e.g., low-SES college students, young people of color, members of the LGBTQIA+ community, people with disabilities, people with immigrant backgrounds) have supported dissemination efforts by speaking directly about their own experiences of engagement with PJP [e.g., 26,27]. Recruitment of participants started

Upon joining, participants received a study information sheet (available in English and Spanish; both versions included in the data repository), and electronically consented to participation. They provided a single piece of contact information — an email address or mobile phone number — which was used to distribute weekly invitations to participate. (Contact information has been stripped from the dataset.)

One key decision at the project outset involved the matter of sampling — or, in plain terms, the question of who could participate in PJP. Our decision reflected the need to bridge a scientific orientation to systematic sampling and comprehensive data collection, our human rights-based commitment to democratizing the process of knowledge production, and a humanities-oriented recognition that each individual voice has value in its own right. Rather than attempt to create a sample that was statistically representative of any particular population, we elected instead to welcome any interested participant and capture as broad a range of perspectives as possible, while also taking measures to include voices that might not otherwise be preserved in the historical record. [27]

We took this more expansive approach for several reasons. First, while the project took care not to promise any sort of therapeutic benefit, we were aware of the evidence that some people experience mental health benefits from journaling and other forms of creative expression and reflection. [28,29] In a moment of global turmoil, we thus saw PJP as an opportunity to leverage our research skills and experience in a way that might yield some societal benefit. Rather than limit access to a potentially meaningful resource in such trying times, we elected instead to make PJP open and accessible to anyone interested — including teenagers aged 15–17, with parental/guardian consent.

A second reason for our expansive approach to recruitment derives from the interdisciplinary nature of PJP and the recognition that different kinds of data can support different types of analyses. PJP was not designed to produce generalizable results, but rather to create a robust dataset that could support a range of analyses that would not be possible though quantitative approaches alone, as we elaborate below.

## A mixed-methods approach

To support this goal, we developed a comprehensive baseline quantitative survey that participants completed in Week 1 before they began creating journal entries. Working in collaboration with political scientist Abigail Fisher Williamson, we designed a baseline survey that employed a combination of validated and original survey items on a wide range of topics including demographics (e.g., age, gender, income, country of residence, etc.), political leanings, media consumption patterns, insurance status, self-reported physical and mental health status, COVID-19 exposure, COVID-related precautions, and loneliness/social isolation. Validated survey items were drawn, or in a few instances adapted, from a range of domain-specific sources on demographics (e.g., Cooperative Congressional Election Study (CCES)), individual and collective views on health (e.g., American Health Values Survey (AHVS)), subjective experiences of mental health (e.g., the DeJong Gierveld Loneliness Scale and the PROMIS Scale), and COVID-related exposures and practices (e.g., Epidemic-Pandemic Impacts Inventory (EPII)). [30–34] As noted earlier, several sets of questions, including biweekly physical and mental health questions, were then repeated periodically, yielding quantitative measures of change over time that can be analyzed in conjunction with participants' qualitative entries.

Using these data to filter and generate subsets, researchers can analyze participants' narrative writing to illuminate the lived experiences of specific demographic or professional groups, such as new mothers, [35] health care providers, [36] or college students. [37] Similarly, longitudinal journals make it possible to explore how participants' perspectives — as individuals and as members of multiple, overlapping demographic groups (e.g., retirees in the Northeastern US who live alone, or essential workers who are single parents living in urban areas) — changed over time.

## Data collection

All data were collected using the Qualtrics survey platform. The core of PJP-1 involved weekly opportunities to create journal entries in the language and format of their choice (text, image, and/or audio). After joining the project and completing

the baseline survey, participants were then invited to create two journal entries in response to suggested narrative prompts. [38] In each subsequent week, participants received weekly invitations, via their choice of email or SMS (text message), to create new entries in their journals. Each invitation included a link to that week's journaling prompts and accompanying survey questions. [39,40] The first journaling prompt was the same every week: "How is the coronavirus pandemic affecting your life right now?" The second entry involved a choice of two prompts, typically including one focused on subjective experience (e.g., "Think about the people closest to you. How has the pandemic affected them lately?") and another with an external focus (e.g., "How has the pandemic affected your view of government and its role in your life?").

Participants could join the project at any point between May 2020 and May 2022. PJP-1 employed a cohort design. Regardless of when they joined, all participants received the narrative prompts and accompanying survey questions in the same order (i.e., Week 1 questions followed by Week 2, etc.). Importantly, participants could stop participating at any point — or they could stop participating for a time and later restart. Retention was encouraged with a monthly raffle of three $100 gift cards. All individuals who had contributed in a given month were eligible for that month's raffle.

## Materials

The dataset housed at QDR contains all qualitative journal entries and quantitative survey responses submitted to both the English and Spanish versions of the PJP-1 platform. An accompanying set of documents is included to help users understand the design, scope, implementation, and public-facing dimensions of PJP. A full table of contents for the project can be found in Fig 1.[see also 41]

The accompanying materials are organized in two groupings. The first set of documents (A. PJP-1 Descriptive Materials) includes all survey items and narrative prompts, including both the English and Spanish versions. A survey planning document enumerates the order in which survey items and narrative prompts were deployed, including the subset of survey items that were asked on repeated occasions. Other materials include a glossary of key terms, IRB-approved consent documents, and other explanatory materials. The second set (B. PJP Background Materials) includes documents outlining the personnel involved (key research personnel, project advisors, and student advisors), funding and sponsorship details, and a description of subsequent spin-off projects that were developed using adapted forms of the PJP platform. This set also includes full documentation of PJP's public-facing materials, including archived copies of all pages of the project website and screenshots of social media accounts (Instagram, Facebook, and Twitter).

## Data curation

The PJP team consulted with staff at QDR early on. Initial conversations focused on appropriate consent to allow sharing and archiving the data [42] and the general scope and volume of data to expect. In semi-regular consultations over the next few years, PJP and QDR settled issues of file naming, file organization, and the types of accompanying documentation to include. The full dataset includes over 3,200 files (16 documentation files, 23 files of background materials, 4 tabular datasets in Excel format, and 3,237 multi-media files shared by participants) that are systematically named and can be linked to individual survey responses. Following best practices in data archiving [43], the data on QDR's servers are tightly secured against unauthorized access as well as accidental data changes or loss. The dataset as deposited forms a permanent record of the projects results. In the unlikely case that future changes of the data are necessary (e.g., due to copyright claims by third parties on included materials), the repository will create a new version of the dataset and clearly outline any changes. To further safeguard the data, any access to the archive in the next 25 years will run through a careful data access protocol as outlined below.

In preparing contributed materials for publication, QDR screened for duplicate and invalid files and ensured tabular data and multi-media files matched. PJP researchers and QDR consulted closely on the eventual structure of the project, ensuring that the published data meet both PJP's vision for the project and best standards for data archiving. [44] Finally,

| FILENAME | FORMAT | SUBSECTIONS (tabs/folders/files) | PUBLIC OR RESTRICTED? |
|---|---|---|---|
| **PJP_A_DESCRIPTIVE MATERIALS** | | | |
| PJP_A-01_DataNarrative_Phase1.pdf | PDF | | public |
| PJP_A-02_TableOfContents.xlsx | spreadsheet | | public |
| PJP_A-03_TermsOfAccess.pdf | PDF | | public |
| PJP_A-04_GlossaryOfKeyTerms.pdf | PDF | | public |
| PJP_A-05_ColumnLabels.xlsx | spreadsheet | | public |
| PJP_A-06_QualitativeNarrativePrompts.xlsx | spreadsheet | | public |
| PJP_A-07_QuantitativeSurveyQuestions.xlsx | spreadsheet | 1. Overview<br>2. Baseline Questions_ENGLISH<br>3. Baseline Questions_SPANISH<br>4. Follow-Up Survey Questions_ENGLISH<br>5. Follow-Up Survey Questions_SPANISH | public |
| PJP_A-08_SurveyPlanningSchedule.xlsx | spreadsheet | | public |
| PJP_A-09_ParticipantInformationSheet_EN.pdf | PDF | | public |
| PJP_A-10_ParticipantInformationSheet_SP.pdf | PDF | | public |
| PJP_A-11_AudioCatalog_PUBLIC.xlsx | spreadsheet | | public |
| PJP_A-12_PrelimaryTagList_AUDIO.pdf | PDF | | public |
| PJP_A-13_ImageCatalog_PUBLIC.xlsx | spreadsheet | | public |
| PJP_A-14_PreliminaryTagList_IMAGES.pdf | PDF | | public |
| PJP_A-15_EditAndErrorLog.xlsx | spreadsheet | | restricted |
| **PJP_B_BACKGROUNDMATERIALS** | | | |
| PJP_B-1_ResearchPersonnelAndAdvisors.pdf | PDF | | public |
| PJP_B-2_Sponsors.pdf | PDF | | public |
| PJP_B-3_SubsequentPJPProjectsAndSponsors.pdf | PDF | | public |
| PJP_B-4_Websites | folders containing HTML files | PJP_B-4a_MainWebsite<br>PJP_B-4b_MyJournal-FeaturedEntriesWebsite<br>PJP_B-4c_PicturingThePandemic_ExhibitionWebsite | public |
| PJP_B-5_SocialMediaSnapshots | folder containing HTML files | PJP_B-5a_FacebookPage_PartialSnapshot.html<br>PJP_B-5b_InstagramPage_PartialSnapshot.html<br>PJP_B-5c_TwitterPage_PartialSnapshot.html | public |
| **PJP_C_DataFiles** | | | |
| PJP_C-1_FullData.xlsx | spreadsheet | 1. Full Data_ENGLISH<br>2. Full Data_SPANISH | restricted |
| PJP_C-2_AudioCatalog_RESTRICTED.xlsx | spreadsheet | | restricted |
| PJP_C-3a_ImageCatalog_RESTRICTED.xlsx | spreadsheet | | restricted |
| PJP_C-3b_ImageCatalog_RESTRICTED_WithThumbnails.xlsx | spreadsheet | | restricted |
| PJP_C-4_AudioFiles | folder | | restricted |
| PJP_C-5_ImageFiles | folder | | restricted |

**Fig 1. Table of Contents to PJP-1 Dataset at QDR.**

repository and researchers agreed on a workflow for handling data requests that reduces barriers for re-use while meeting the access conditions required by PJP. This model of early contact and sustained consultation between researchers and data repository and curators was essential for the successful realization of the long-term archiving of the project. We hope it can serve as a model for other large, complex qualitative and mixed-methods projects seeking to archive their data.

## The dataset

As noted earlier, the PJP-1 dataset includes individual journal entries and accompanying quantitative survey responses from more than 1,800 participants hailing from 55 countries. Of nearly 27,000 journal entries in total, over 2,700 include

images and over 300 are audio files. Below we provide a top-line overview of participant demographics, followed by a brief discussion of patterns of participation and attrition.

### Participant demographics

In total, 1,839 individuals joined the project between May 2020 and May 2022. Of this total, 92 percent (n = 1,692) joined via the English-language platform and the remaining 8 percent (n = 147) joined via the Spanish platform. While the interface itself was offered only in these two languages, participants could create journal entries in any language they wished. As a result, the dataset includes small sets of materials in Chinese, Bangla, and Portuguese in addition to English and Spanish.

As evident in Fig 1, the baseline survey yielded a rich array of quantitative insights into the demographics of individual participants. Participants could decline to answer questions, and some skipped one or more. For the most part, however, participants responded to most of the questions posed.

Nearly 80 percent of all participants (n = 1,460; 79%) identified as women, 17 percent (n = 319) men, and two percent (n = 39) identified with another gender category. In terms of age, nearly half (n = 875) were between 15 and 29 years old, although participants ranged widely in age (see Fig 2). Nearly one-quarter were in their teens (n = 428; 23%); almost one-quarter were in their 20s (n = 447; 24%); just over one-quarter were in their 30s or 40s (n = 462; 26%); and over one-quarter were 50 or older (n = 483; 27%). The youngest participant was born in 2006, and the oldest in 1931.

In terms of highest level of educational attainment, nearly one-quarter of participants reported some college experience (n = 417; 23%); over one-quarter held either an associate's or a bachelor's degree (487; 27%); and a full 35% (n = 643) held a postgraduate degree. Of the remaining 15%, nearly all (n = 247; 14%) had completed high school or a technical or vocational school, and just 1 percent (n = 26) had not completed a high school education. Given the high proportion of student participants (n = 497; 27%), it is likely that many of those reporting lower educational attainment were on a path toward entering the next category of educational attainment (and may have done so by the time the study was complete).

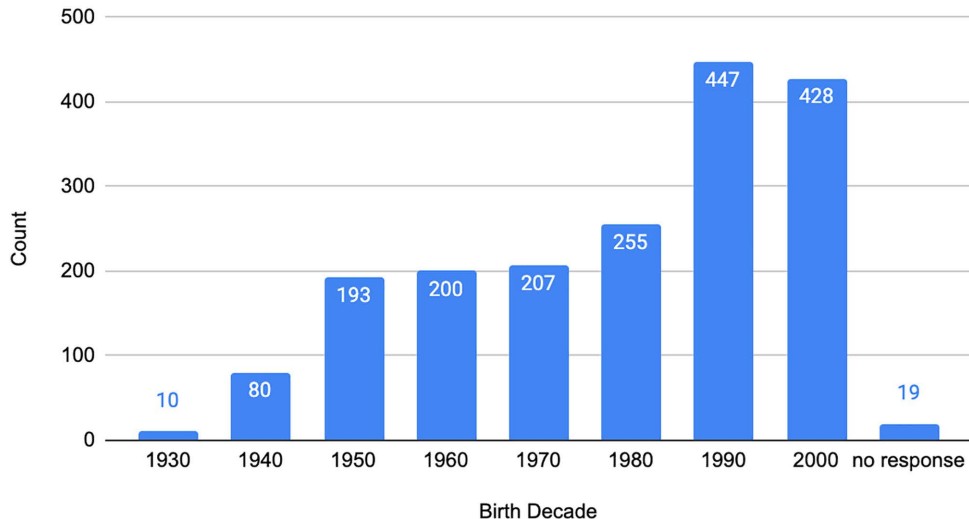

**Fig 2. PJP participants, by birth decade.**

Participants were asked to indicate their annual household income in US dollars. Of those who provided their household income (n = 1,227), three roughly equal groups reported a household income of less than $50,000 (n = 354; 29%), $50-$100K (n = 351; 29%), and $100-200K (n = 344; 28%). The remaining 25 percent reported income of over $200K.

In terms of marital status, over half of participants were single (n = 958; 52%), and nearly 40% were either married (n = 601; 33%) or in a domestic partnership (n = 112; 6%). Smaller numbers reported being divorced (n = 93; 5%), widowed (n = 35; 2%), or another, unspecified status (n = 40; 2%).

Gathering data on participants' racial and/or ethnic background posed one notable challenge. We wanted the interface to be usable and relevant to participants around the globe, yet the racial/ethnic categories typically used in the US are not universally relevant or comprehensible, nor do they translate well into languages other than English. To address this challenge, we asked about racial/ethnic background in two different ways. First, all participants who listed the US as their country of residence were presented with a closed-ended question based on US census categories that allowed respondents to mark more than one category, if relevant. In addition, all participants, regardless of country of residence, were offered an opportunity to name the "racial or ethnic group(s) that best describes you" in a separate, write-in question. Of US-based participants who responded to the first question (n = 1,441; 78%), almost half described themselves solely as White (n = 907; 49%). Eight percent identified as Hispanic/Latinx (n = 151), and six percent identified respectively as Black (n = 102) or Asian/Pacific Islander (n = 111). Nine percent (n = 170) identified with two or more racial/ethnic categories, or as something else.

These demographics can be interpreted in various ways. One might be inclined to focus on the heavy representation of White participants among those living in the US. From another angle, however, it is noteworthy that nearly 30% (n = 534; 29%) identified with a category other than White. In addition, and perhaps most importantly, scholars interested in analyzing the qualitative journal entries will find relatively large numbers of participants in each of the non-White US census categories.

Participants were asked to share their employment status and, if relevant, to indicate whether they were employed as an "essential worker." As noted earlier, over one-quarter (n = 497; 27%) were students (see Fig 3). Of non-students, over half were employed either full time (n = 679; 37%) or part time (n = 315; 17%), with much smaller numbers reporting being homemakers (n = 38; 2%), unemployed (n = 59; 3%), or temporarily laid off (n = 10; 0.5%).

Nearly one-quarter of participants characterized themselves as essential workers (n = 444; 24%; see Fig 4). Of the essential workers who specified their type of work, the most common categories involved health care and other forms of

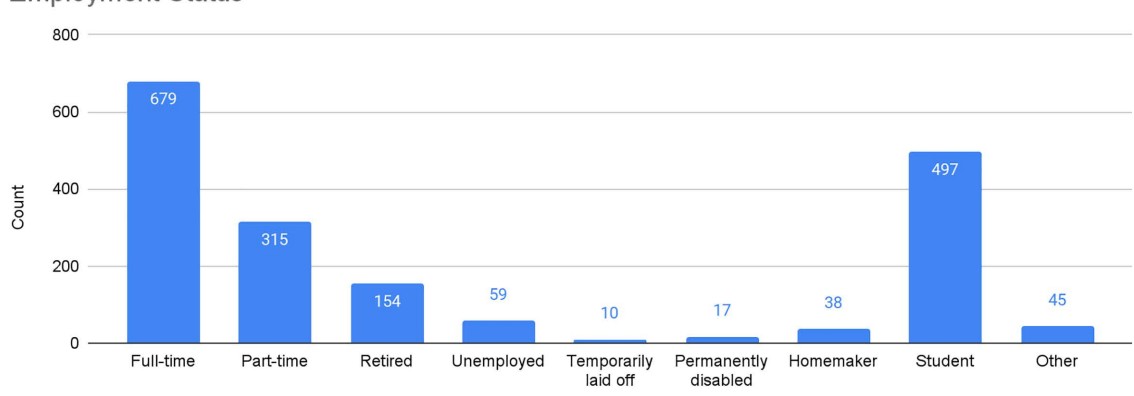

**Fig 3. PJP participants, by employment status.**

## Essential Workers

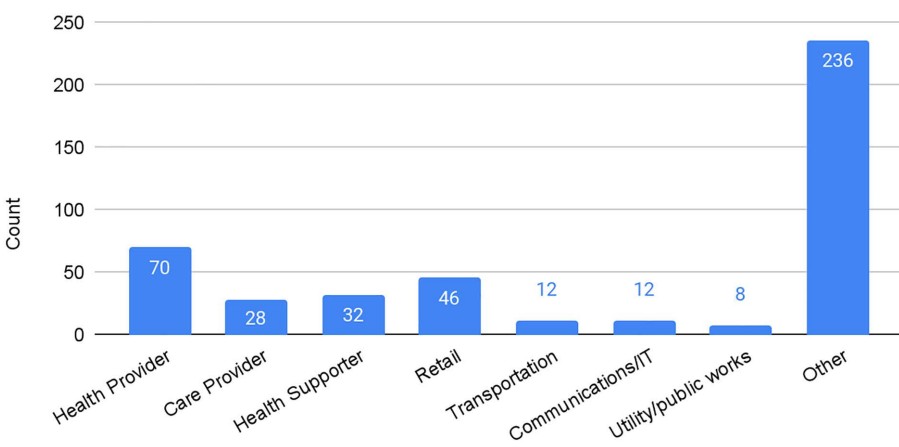

Fig 4. Essential workers among PJP participants, by employment category.

care provision (n = 130) or retail (n = 46), with smaller numbers working in transportation (n = 12), communications/information technology (n = 12), or utilities and public works (n = 8).

Geographically, participants hailed from 55 countries, with the vast majority coming from the United States (n = 1,445), Mexico (n = 85), and Canada (n = 72), and smaller numbers from the Germany (n = 16), United Kingdom (n = 15), China (n = 13), Brazil (n = 13), and elsewhere (see Fig 5). Within the US, participants hailed from 46 states plus Puerto Rico. Best represented regions included the Northeast, Midwest, and West Coast, and best represented states included New York (n = 335), Connecticut (n = 130), and California (n = 127), followed by Michigan (n = 77), Massachusetts (n = 74), and Pennsylvania (n = 54) (see Fig 6) Maps were generates using the "maps" R-package [45], relying on public domain data from Natural Earth [46] (Fig 5) and the US Census Bureau [47] (Fig 6).

Given our recruitment strategy, we would not expect the demographics of the US sample in PJP-1 and the US population to match. Nevertheless, an understanding of how the PJP-1 sample differs from the US population is useful in interpreting PJP-based findings. The PJP sample skews heavily towards the Northeastern US, where the PJP team is located. PJP was successful in recruiting participants from diverse socioeconomic statuses, broadly matching the US population. Despite significant efforts to increase the participation of non-White populations, they remain under-represented in PJP-1, although their rates of participation are higher than in common convenience samples, including those deemed comparatively diverse, such as Amazon MTurk workers. [48] Perhaps most striking is the stark gender imbalance among PJP-1 participants. PJP is not the first qualitative study to face challenges in recruiting male participants. [49,50]

This pattern of participation raises multiple questions about the nature and implications of creating, or in this case not creating, a sample that meets formal criteria of representativeness. Since PJP was not designed to generate a representative sample, we cannot make strong claims of generalizability in relation to any specific population or subpopulation. A dataset of this size and breadth can, however, help identify notable patterns and trends, and it can help generate hypotheses that might be tested using quantitative methods with representative samples. Moreover, while the PJP-1 dataset cannot help discern strong trends or patterns involving groups that are poorly represented in the sample (for instance, cross-national trends or patterns among groups thinly represented in terms of age, disability status, etc.), many groups are sufficiently represented to help in such efforts (e.g., college students, women with children, retirees, etc.). Opportunities for analysis are further amplified by the depth and richness of the qualitative data included in the dataset.

PJP Participants by Country (Greyscale)

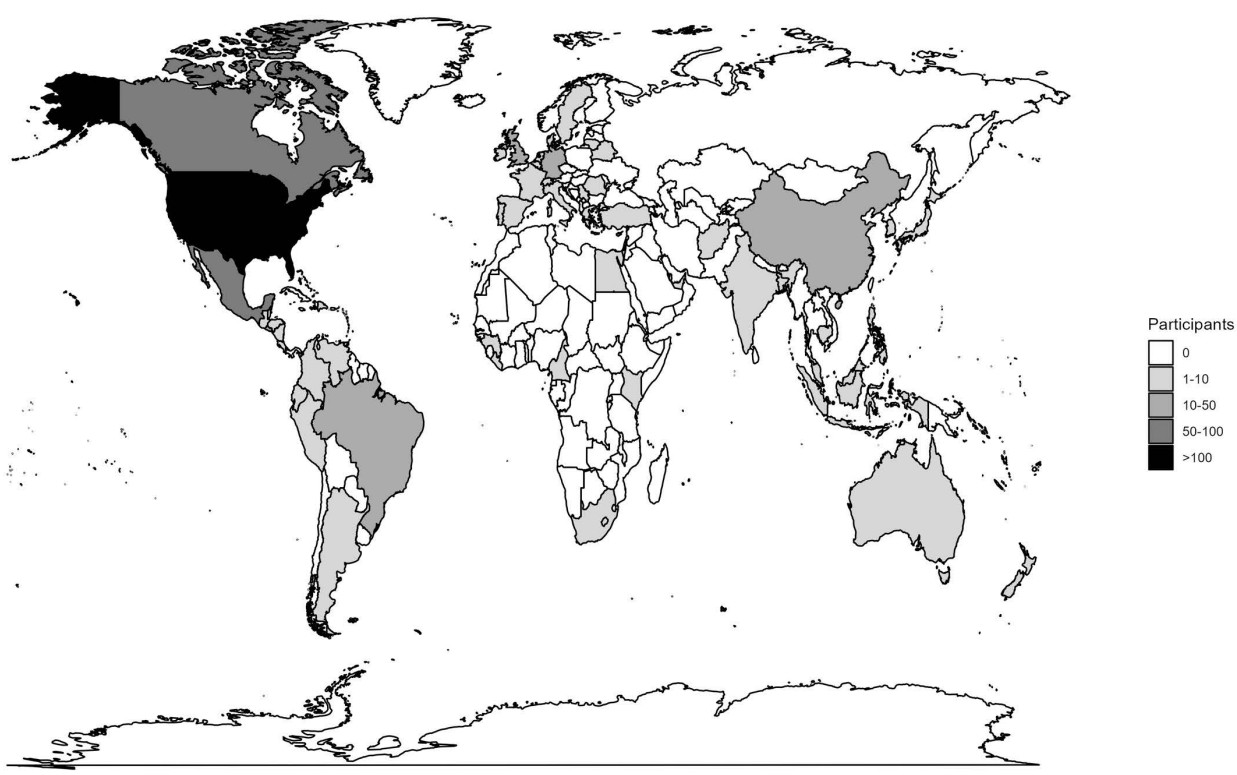

**Fig 5. PJP participants, by country of residence.** The spatial data used to construct the map were obtained from the following open access sources (via the R 'maps' package): Fig 5: Natural Earth, "Free vector and raster map data," https://www.naturalearthdata.com/downloads/, released as public domain data; Fig 6; US Department of Commerce, Census Bureau, "County Boundary File," https://www.census.gov/geographies/mapping-files/time-series/geo/carto-boundary-file.html, released as public domain data.

## Patterns of participation and attrition

Participants' motivations for joining PJP varied widely, as did their patterns of participation. Some joined during the earliest weeks of the project in late spring 2020, but new participants continued to join through mid-spring 2022, when the project was already drawing to a close. Numbers of new participants were highest in the first several months, then in spring 2021 after a period of significant media coverage in the US and beyond. [51] Since participation each week was optional, many participants chose to skip weeks, sometimes refraining from contributing for weeks or even months, then participating again when they felt inclined to respond to PJP's weekly invitations to participate. As a result, both the length of participant journals and the total number of weeks of participation vary widely. Some journals include entries from a limited number of weeks, while others include a steady pattern of contributions over an extended period of time. The largest number of participants (n = 1,839) contributed on at least one occasion. Given the project's cohort design, all participants received the same questions in the same order, regardless of when they joined (i.e., Week 1 followed by Week 2, Week 3, etc.). Nearly half of those who participated in Week 1 participated again in Week 2 (n = 877), with participation rates decreasing over time. By Week 4, the initial number of participants had dropped to approximately one-third of the original number (n = 622). At Week 24, nearly 150 people contributed to their journals (n = 144). At Week 52, a full year after joining the project, just over 100 contributed in (n = 103). By Week 72, the number of participants contributing had dropped to 30, then again to

US-Based PJP Participants by State (Greyscale)

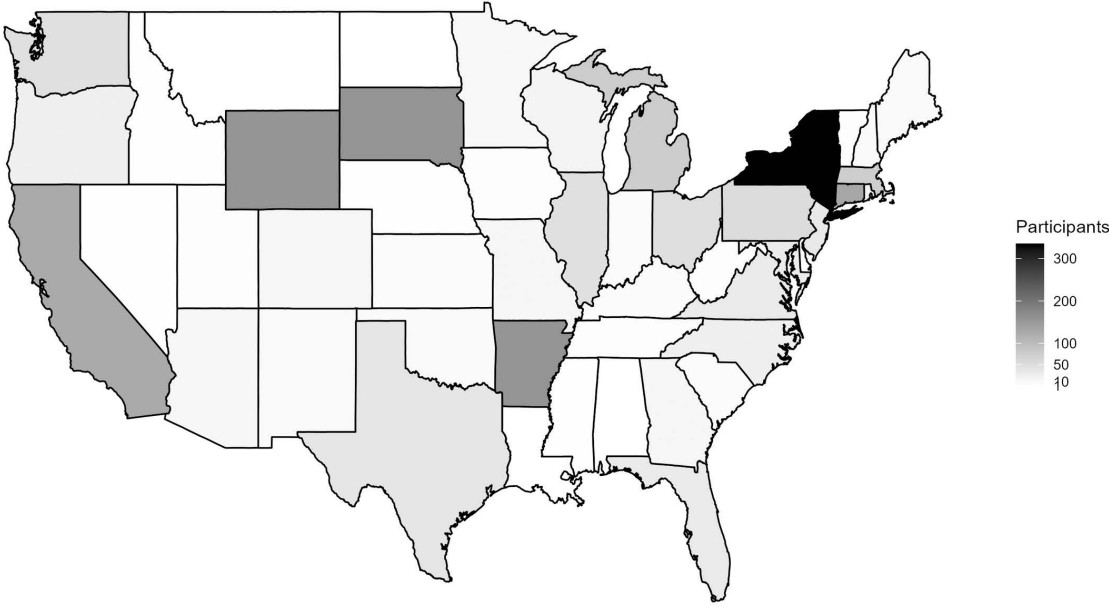

**Fig 6. US-Based PJP participants, by state of residence.** The spatial data used to construct the map were obtained from the following open access sources (via the R 'maps' package): Fig 5: Natural Earth, "Free vector and raster map data," https://www.naturalearthdata.com/downloads/, released as public domain data; Fig 6; US Department of Commerce, Census Bureau, "County Boundary File," https://www.census.gov/geographies/mapping-files/time-series/geo/carto-boundary-file.html, released as public domain data.

half that number by Week 104, representing the small number of individuals who joined at the very beginning and stuck with the project until the end of its first phase (see Fig 7).

Four key factors help explain the variation we see in the dataset. First, many people who participated on one occasion were uninterested in continuing to participate, which we recognized and accepted. Participants could always request to be removed from our distribution list, or simply ignore invitations to participate. Second, given the project's cohort design, late starters had fewer overall opportunities to contribute than those who joined earlier. For example, someone who joined in late May 2021, one year from PJP's launch, created their Week 1 journal entries at the same time that those who joined in May 2020 were completing Week 53. Third, the Spanish-language interface launched several weeks after the English-language interface. As a result, only English-speaking participants who joined in May 2020 had the maximum number of 105 weekly opportunities to contribute. Finally, since individuals could skip weeks, some journalers contributed on fewer occasions than the final week number in which they contributed. For instance, an individual contributing in Week 40 may have skipped five weeks, thereby contributing on just 35 rather than 40 occasions.

## Results and analyses to date

Findings from the PJP-1 dataset have already begun to advance understanding of the impact of the COVID-19 pandemic on various areas of individual and collective health and well-being. Thus far, findings from analyses of PJP-1 data have focused largely on mental health and well-being, with additional analyses focusing on women's health and public health.

These studies focus on subsets of journals, or subsets of journal entries, rather than the full dataset. For example, some analyses focus on all journals from people who belong to a specific demographic group, or whose journals employ specific keywords. In cases where a search by demographic categories or keywords yields an especially large dataset, we

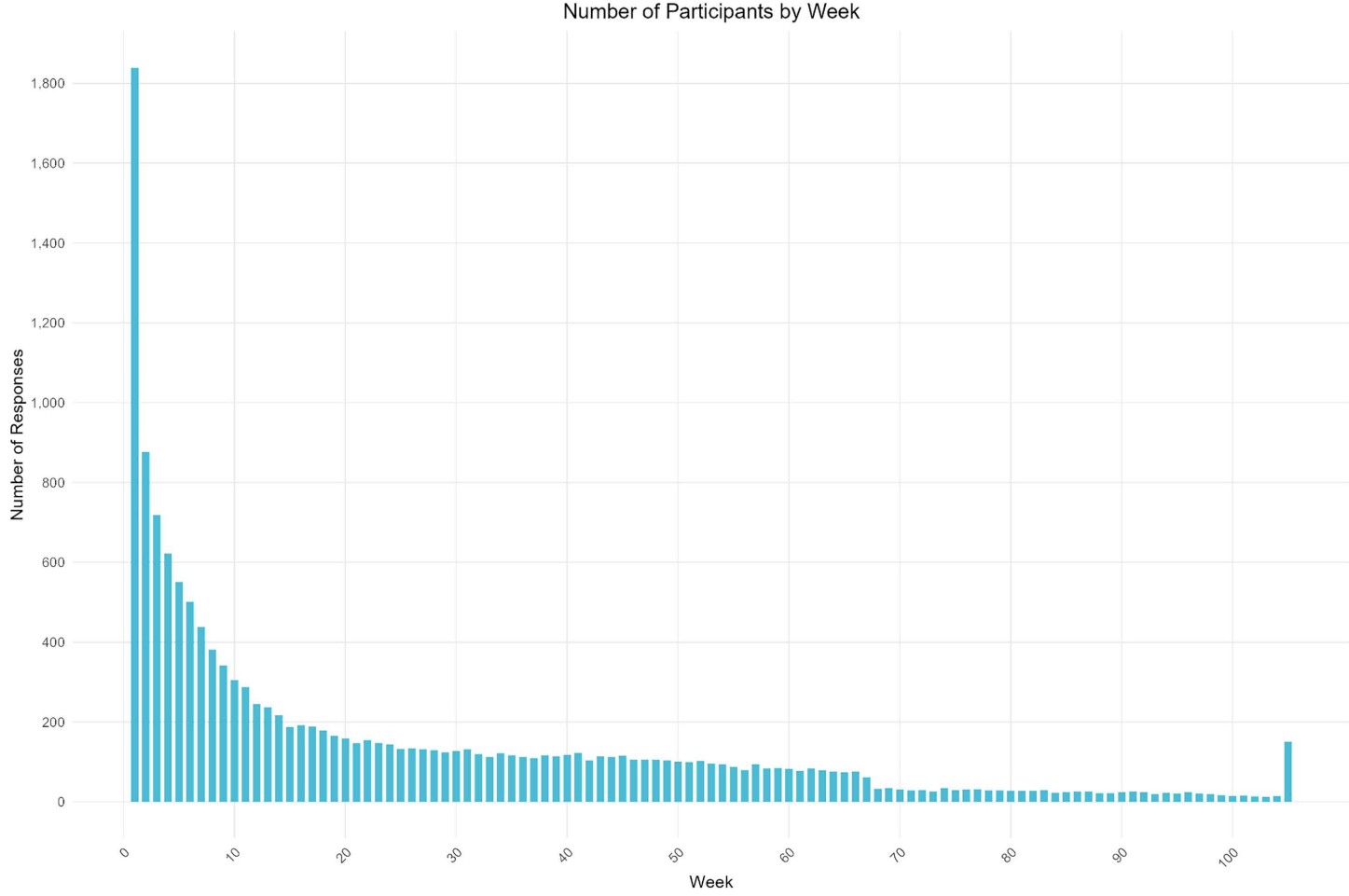

**Fig 7. Patterns of participation and attrition among PJP participants.** The final invitation to contribute (in Week 105) was sent to all participants regardless of when they had joined, which explains the higher response rate.

have extracted a random sample for analysis. Below we summarize these early analyses, noting that future researchers will be able to approach the dataset with fresh eyes and new research questions.

From a mental health standpoint, many aspects of the pandemic created, or exacerbated, stress and anxiety that found expression in PJP-1 journals. Widespread experiences included fear of exposure to a poorly understood virus that could cause severe illness or death; fear that loved ones, friends, coworkers, or community members might be exposed; lost work opportunities and resulting loss in income; disruptions to educational opportunities and plans; loss of opportunities for social interaction; and increased risk of both social isolation and loneliness. While these forms of stress and anxiety affected many if not most people, their impact was especially heavy on individuals who already were experiencing serious mental illness, or who were close to others facing serious mental illness.

Although PJP-1 was not created as a mental health intervention, its design was informed by scholarship on the potential mental health benefits of writing and other forms of free and creative expression, as noted above. The first collection of papers to emerge from the study is a special issue focusing on "Journaling and Mental Health" [21,52], in which PJP team members and collaborating scholars analyzed subsets of PJP data to investigate this theme from a variety of angles. Several articles focused on the impact of the pandemic on specific groups, among them Black women with caregiving

obligations, [53] health care providers, [36] new mothers, [35] and community college students in New York City [54]. Other articles consider how pandemic restrictions generated — or exacerbated — feelings of loneliness,[55,see also 56] and how a popular conception of "languishing" came to capture some people's pandemic-era experiences of stagnancy and isolation. [57]

In a second collection of articles ("Student Experiences of Covid-19 around the Globe" [37]), PJP team members and collaborators home in on the experiences of students and, to a lesser extent, educators, many of whose mental health and well-being were deeply affected by the pandemic. Contributions explore how the pandemic interrupted plans, educational opportunities, and everyday life for students at a range of educational levels (high school to graduate school) in a range of countries around the globe, including the US, [20,58] Mexico, [59,60] Brazil, [61] China, [62] and South Africa. [63]

Another area of focus in our initial analyses involves women's health. Publications to date explore women's experiences of caring for children and other loved ones in the context of rapidly shifting household dynamics, altered work arrangements, prolonged isolation from extended family and friends, and decreased access to public and private institutions (e.g., schools, child-care centers). Several analyses explore how cultural expectations — for instance, of femininity, motherhood, and kinship — have been embraced, resisted, and reconfigured as women of different backgrounds struggle to adapt their current lives and family-building aspirations to new pandemic realities. [64–66] Some mourn disruptions to key life events surrounding childbirth and child raising, while others face distinct challenges and opportunities. In all instances, women's experiences of health, well-being, and reproductive decision-making are informed by their intersecting social positionalities, including such factors as age, race/ethnicity, educational attainment, and socioeconomic status. In another analysis, journals by Black women in the US reveal the important role of social connectedness in coping with adversity, particularly in light of historical expectations of Black women to be pillars of strength within their families despite exposure to persistent structural vulnerability [53].

Third, the PJP-1 dataset already has begun to generate valuable insights for researchers and community-based providers in public health. In one article, for instance, we explore the contentious issue of vaccine uptake. [67] We draw on longitudinal journals from a subset of PJP-1 journalers to explore the worries, hesitations, and anger that white, seemingly vaccine-compliant women experienced in the months soon after COVID-19 vaccines became available. Through examination of the moral "self-talk" evident in their journals, we found that these women deeply distrusted both their institutions and their fellow citizens, and that they expressed this distrust in part through demonization of others' vaccine choices and defensiveness about their own choices. In another study, PJP-1 findings led us to predict that journaling might be a valuable research method for learning about the enduring impact of COVID-19 on immigrant women in New York City and, moreover, about the impact of community-based organizations' (CBOs') efforts to offer psychosocial support and strengthen community-building. [64,65] Using a study design involving journaling, focus groups, and in-depth interviews, we learned how immigrant women in NYC have confronted the pandemic, in addition to gathering preliminary evidence that journaling-based projects may be an effective way for CBOs to provide psychosocial support in community settings.

The PJP team has deposited PJP-1 data at QDR so that other scholars can draw upon our rich data to explore additional themes and topics, both in the near-term and well into the future. For the same reasons, data from several of our spin-off studies will be deposited at QDR as well.

## Conclusions: Implications and future possibilities

In this article, we have introduced an innovative, mixed-methods dataset that was generated by leveraging digital technology to invite ordinary people around the world to document the impact of the COVID-19 pandemic on their everyday lives.

The PJP-1 dataset, which was produced between May 2020-May 2022, already has generated important insights into a range of issues, including the impact of the COVID-19 pandemic on mental health, women's health and fertility decision-making, and vaccine-related hesitancy, among other topics. These studies signal only a few of the many research possibilities involving this dataset. Below we highlight a number of future possibilities, then point to some of the broader implications of the study overall.

## Promising avenues for thematic analyses

One way to think about the value of the PJP-1 dataset is in thematic terms. While mental health has already been the focus of a range of PJP-1 publications to date, the impact of the COVID-19 on individual and collective mental health has been profound and far-reaching, and given its prevalence in PJP-1 journals, much more can be learned. Similarly, more can be learned about the impact of the pandemic on access to reproductive health care, fertility decision-making, and infertility treatment. PJP-1 journals can also support studies of how specific groups experienced the first two years of the pandemic, including people of all ages who are living with disabilities, caregivers for people living with disabilities, and members of LGBTQ+ communities.

In terms of health care institutions and infrastructure, another important theme involves the ways in which individuals and families endured disruptions to care, including routine and preventative care, care for ongoing or chronic conditions, and care for acute conditions as well as emergency situations.

From a public health standpoint, the PJP-1 dataset can generate insights into a wide range of topics, including public perceptions of emergency preparedness efforts; uptake of precautionary measures, including but not limited to vaccination; and more population-specific topics such as parents' decision-making in relation to protecting their children's health. Additionally, PJP-1 journals can help shed light on the ways in which public health messaging can strengthen or weaken public trust in leaders, government institutions, and public health science more generally. Such analyses may be especially valuable in countries where early public health messaging efforts around COVID-19 involved key missteps and failures, including the US.

Debates about fairness, equity, health-related deservingness, [68,69] and related moral questions are also key themes in the PJP-1 dataset. In addition to documenting their own struggles to protect their health and the health of those closest to them, many participants also documented their deliberations regarding key questions — questions of who deserves what and why — in relation to everything from protective equipment and vaccines to medical attention, hospital beds, ventilators, and other forms of physical, material, and social care and support. Qualitative researchers in bioethics, public health ethics, medical anthropology, and other social sciences of health have much to learn from these data.

## Future possibilities

Looking forward, some of the PJP-1 dataset's more unusual features invite creative thinking about new possibilities for qualitative and mixed-methods research. First, PJP was designed quickly in response to an unfolding global health crisis, suggesting that researchers can, if need be, work quickly — in both collaborative terms and in dialogue with IRBs — to create meaningful research studies that can document, and potentially analyze, major events in real time.

Second, PJP's innovative approach to data collection stands as an example for researchers keen on using mixed-methods approaches, especially those that leverage digital technology to provide participants opportunities to generate data in ways that match their everyday habits and practices. Unlike survey-based research studies that employ mostly closed-ended questions, PJP's journaling platform emphasized open-ended submissions and successfully encouraged detailed responses. It invited responses to open-ended questions using any combination of text, images, and/or audio material. In these respects, and also by facilitating participation either using a computer or a smartphone (or other electronic device), PJP-1 sought to give participants as much flexibility as possible in choosing how, when, in what medium, and with what frequency they would participate. This approach to collecting in-depth, qualitative data at scale makes PJP part of a new current of large-n, qualitative work [70].

Third, PJP invites reflection on the value, merits, and challenges of working with non-representative research samples. While this facet of the sample does limit the generalizability of findings in some respects, it offers one model for confronting the challenges involved in democratizing access to the research enterprise, and to the process of knowledge production. Fundamental to the PJP model is a commitment to offering a meaningful, substantive benefit to research participants that can motivate participation and lead participants to provide data whose validity is strengthened by the

fact that it is meant to hold personal value and is not simply produced for researchers' benefit. In addition, it opens up promising pathways for conducting large-n research at a time in which quantitative research mechanisms are increasingly coming into question, for instance because of low response rates (to online and phone surveys) [71,72] and increasing concerns that tools like MTurk are decreasing in reliability and validity due to responses from AI bots rather than human participants [73,74].

Fourth, the PJP team has sought opportunities to involve research participants in exhibiting and analyzing study findings, for instance by co-organizing a public forum in partnership with a state-level legislative commission ("Journaling in COVID Times: A Roundtable on Storytelling, History, Equity, and Mental Health"); prevailing on a leading national scholarly association to permit research participants to join study PIs on a conference panel ("Grassroots Collaborative Ethnography: Insights from the Pandemic Journaling Project"); and partnering with contributors to generate the international multimedia exhibition mentioned earlier (*Picturing the Pandemic: Images from the Pandemic Journaling Project*), with five sites in four countries.

In addition, we have developed several major spin-off research projects that center participant involvement. First, data collection is now underway for a longitudinal study of the impact of the pandemic on first-generation college students and their families in the US, for which first-generation students are core members of the research team. In a second spin-off project, we are working with community-based organizations in New York City to explore the enduring impact of the pandemic on immigrant women in the Latinx and South Asian communities and offer recommendations on how better to meet the needs of these communities.

Overall, we hope that researchers both now and in future generations will benefit from analyzing aspects of the PJP-1 dataset as they continue grappling to understand the enduring impact of the COVID-19 pandemic on the health, everyday lives, and life trajectories of ordinary people in North America and around the world.

## Acknowledgments

The authors wish to express their gratitude to all participants in the Pandemic Journaling Project; to Abigail Fisher Williamson and Alice Larotonda for their vital contributions in creating PJP; to past and present members of the PJP Core Team, including Andrea Flores, Clare Wang, Salma Mutwafy, Jolee Fernandez, Nathalie Peña, Ana Perez, Sofia Boracci, Alice Jo, Lauren Deal, Imari Smith, Emily Nguyen, Becca Wang, and Kiran McCloskey; and to Dessislava Kirilova at the Qualitative Data Repository.

## Author contributions

**Conceptualization:** Sarah S. Willen, Katherine A. Mason.

**Data curation:** Sarah S. Willen, Heather M. Wurtz, Sebastian Karcher.

**Funding acquisition:** Sarah S. Willen, Katherine A. Mason.

**Investigation:** Sarah S. Willen, Katherine A. Mason, Heather M. Wurtz.

**Methodology:** Sarah S. Willen, Katherine A. Mason, Sebastian Karcher.

**Project administration:** Sarah S. Willen, Katherine A. Mason, Heather M. Wurtz.

**Writing – original draft:** Sarah S. Willen.

**Writing – review & editing:** Sarah S. Willen, Katherine A. Mason, Heather M. Wurtz, Sebastian Karcher.

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
