## [Decision Letter · Decision Letter 0]

24 Apr 2025

PONE-D-25-02185
The Pandemic Journaling Project: A new dataset of first-person accounts of the COVID-19 pandemic
PLOS ONE

Dear Dr.  Karcher,

Thank you for submitting your manuscript to PLOS ONE. After careful consideration, we feel that it has merit but does not fully meet PLOS ONE’s publication criteria as it currently stands. Therefore, we invite you to submit a revised version of the manuscript that addresses the points raised during the review process.

We look forward to receiving your revised manuscript.

Kind regards,

Benjamin M. Liu, MBBS, PhD, D(ABMM), MB(ASCP)

Academic Editor

PLOS ONE

Journal Requirements:

4. We note that Figures 5 and 6 in your submission contain map/satellite images which may be copyrighted. All PLOS content is published under the Creative Commons Attribution License (CC BY 4.0), which means that the manuscript, images, and Supporting Information files will be freely available online, and any third party is permitted to access, download, copy, distribute, and use these materials in any way, even commercially, with proper attribution. For these reasons, we cannot publish previously copyrighted maps or satellite images created using proprietary data, such as Google software (Google Maps, Street View, and Earth). For more information, see our copyright guidelines: http://journals.plos.org/plosone/s/licenses-and-copyright.

 a. You may seek permission from the original copyright holder of Figures 5 and 6  to publish the content specifically under the CC BY 4.0 license. 

Additional Editor Comments:

Line 82-85: "When the COVID-19 pandemic began spreading rapidly around the globe, experts struggled to understand the epidemiology of the virus; to produce narratives that anxious publics could understand; and to generate practical guidelines that could interrupt viral transmission and minimize morbidity and mortality." There are no references to support this statement. More references should be cited, with the following one as an example (citing is optional):

Liu BM, Yao Q, Cruz-Cosme R, Yarbrough C, Draper K, Suslovic W, Muhammad I, Contes KM, Hillyard DR, Teng S, Tang Q. Genetic Conservation and Diversity of SARS-CoV-2 Envelope Gene Across Variants of Concern. J Med Virol. 2025 Jan;97(1):e70136. doi: 10.1002/jmv.70136. PMID: 39744807.

"As this new virus jumped across borders, countries, and continents, it quickly became clear that different regions, communities, and individuals would be affected in vastly different ways": There are no references to support this statement. More references should be cited, with this one (PMID: 40137747) as an example (citing is optional)

Reviewers' comments:

Reviewer's Responses to Questions

**Comments to the Author**

1. Is the manuscript technically sound, and do the data support the conclusions?

Reviewer #1: Yes

2. Has the statistical analysis been performed appropriately and rigorously? 

Reviewer #1: Yes

3. Have the authors made all data underlying the findings in their manuscript fully available?

Reviewer #1: Yes

4. Is the manuscript presented in an intelligible fashion and written in standard English?

Reviewer #1: Yes

5. Review Comments to the Author

Reviewer #1: The COVID-19 epidemic has changed the lives of people in different regions and backgrounds for a long time, especially their views and corresponding ways on medical, health and other public fields.

Unlike professional research papers in virology, basic medicine, or clinical medicine, this paper focuses more on introducing the methodology of the innovative multimedia dataset of The Pandemic Journalism Project, baseline characteristics of the included population, and prospects for future applications.

Thanks to the advancement of computer technology, human society has ushered in the era of artificial intelligence, and the discourse power of traditional social media has gradually become more accessible to the general public. People have the opportunity to obtain equal access to the possibility of information dissemination. This may enable researchers with different purposes in the future to objectively face the views and choices of individuals in major historical events, as well as the phased results of the real world that have arisen from them.

The paper provides a detailed description of the reasons for the initiation of the dataset, inclusion principles, salient features, goals and objectives, implementation methods, data collection, data security, baseline characteristics of the population, population mobility, promotion and application areas, and future possibilities. The logic is clear and the description is sufficient, which helps people who are interested in this project to understand, participate, and apply it.

But this does not mean that there is no room for discussion in the paper.

1. Although Figure 1 is a very detailed project directory, it needs to be reconsidered whether it is suitable as a figure format for publishing scientific papers. If readers are encountering tables for the first time, although they understand that they are the framework and content of the project, in order to truly understand, they must read the entire paper or operate in the application to have a chance to understand such a rich and comprehensive system document.

2. Figures 2 to 7 are all descriptions of population baseline characteristics. In other words, from line 396 to line 559, such a long content is only a general description of population baselines in scientific research papers. In other words, it is possible that a table can present all the content more concisely. Of course, as a detailed introduction to interdisciplinary writing or projects, this arrangement may not be unreasonable.

3. Traditional research is either prospective or retrospective, but how does this study ensure the authenticity of the records? What does it mean that if someone records documents or videos that do not match reality, how is the project identified? After all, people may have different appearances in front of the camera and in real life.

4. As stated in the paper, the project has already begun to be applied in areas such as population mental health, protection of women's and children's rights, vaccine injection, reproductive decision-making, and even immigration and family impact. How are these data quantitatively applied? For example, how is video information converted into data that can be used for textual research?

5. As an open bidirectional database, how is data security guaranteed? Does it mean that the original creator can change the original record?

6. PLOS authors have the option to publish the peer review history of their article (what does this mean?). If published, this will include your full peer review and any attached files.

Reviewer #1: No

---

## [Author Response · Author response to Decision Letter 1]

21 Jul 2025

Thank you for the helpful comments and the opportunity to revise our manuscript. Below please find our detailed response to editorial and reviewer comments. The uploaded response letter includes a more helpfully formatted version of theses responses.

With best regards,

Sarah Willen, Kate Mason, Heather Wurtz, and Sebastian Karcher

Editor’s Comments and requests

1.Please ensure that your manuscript meets PLOS ONE's style requirements, including those for file naming.

RESPONSE:Thank you for this guidance. We have now modified the manuscript to meet these requirements. In specific, we have 1) removed street addresses from author affiliations; 2) reformatted headers to meet the specifications (i.e., sentence case vs. all caps); and 3) reformatted filenames following template guidelines.

RESPONSE: We are now providing access to two different datasets.

An unrestricted dataset that includes the summary data and analysis code that was used to produce the figures in the paper is available on the Harvard Dataverse at https://dataverse.harvard.edu/previewurl.xhtml?token=d0981257-e6e7-4b8f-a7d8-49d692f25545 (this is a preview URL – the final version will be published at https://doi.org/10.7910/DVN/QTQ3V7).

The full PJP-1 dataset is restricted due to identifiable human participant content under the IRB approval for the study by the UConn IRB. Access to the data can be requested via the “Request Access” button on the dataset’s landing page (https://doi.org/10.5064/F6PXS9ZK) at the Qualitative Data Repository and follows the policy described in the terms of access document (https://doi.org/10.5064/F6PXS9ZK/7UYI4F)

RESPONSE: We have now modified the manuscript to meet these requirements. Changes, including citations for the informed consent materials themselves, can be found on p. 11 of the tracked-changes version of the manuscript.

4. We note that Figures 5 and 6 in your submission contain map/satellite images which may be copyrighted. All PLOS content is published under the Creative Commons Attribution License (CC BY 4.0), which means that the manuscript, images, and Supporting Information files will be freely available online, and any third party is permitted to access, download, copy, distribute, and use these materials in any way, even commercially, with proper attribution. For these reasons, we cannot publish previously copyrighted maps or satellite images created using proprietary data, such as Google software (Google Maps, Street View, and Earth). For more information, see our copyright guidelines: http://journals.plos.org/plosone/s/licenses-and-copyright.

RESPONSE: These figures were generated by us using R so we own any applicable copyright. The full code and data to generate the figures are now included in the reproducibility dataset on the Harvard Dataverse (see above).

Additional Editor Comments:

Line 82-85: "When the COVID-19 pandemic began spreading rapidly around the globe, experts struggled to understand the epidemiology of the virus; to produce narratives that anxious publics could understand; and to generate practical guidelines that could interrupt viral transmission and minimize morbidity and mortality." There are no references to support this statement. More references should be cited, with the following one as an example (citing is optional):

Liu BM, Yao Q, Cruz-Cosme R, Yarbrough C, Draper K, Suslovic W, Muhammad I, Contes KM, Hillyard DR, Teng S, Tang Q. Genetic Conservation and Diversity of SARS-CoV-2 Envelope Gene Across Variants of Concern. J Med Virol. 2025 Jan;97(1):e70136. doi: 10.1002/jmv.70136. PMID: 39744807.

RESPONSE: Supporting references have now been added on p. 4 of the tracked-changes version of the manuscript.

"As this new virus jumped across borders, countries, and continents, it quickly became clear that different regions, communities, and individuals would be affected in vastly different ways": There are no references to support this statement. More references should be cited, with this one (PMID: 40137747) as an example (citing is optional)

RESPONSE: Supporting references have now been added on p. 4 of the tracked-changes version of the manuscript.

Reviewer 1: Comments to the Author

1. Although Figure 1 is a very detailed project directory, it needs to be reconsidered whether it is suitable as a figure format for publishing scientific papers. If readers are encountering tables for the first time, although they understand that they are the framework and content of the project, in order to truly understand, they must read the entire paper or operate in the application to have a chance to understand such a rich and comprehensive system document.

RESPONSE: Thank you for these observations. While we are open to omitting Figure 1, we believe it provides a useful guide for readers interested in understanding both the design of the dataset and the degree of planning and thought invested in ensuring that researchers uninvolved in the original project would understand the data collection process, the contours of the dataset, the differences between publicly accessible and restricted sections of the dataset. Above all, the goal of Figure 1 is to provide researchers potentially interested in working with the dataset a clear sense of what it includes, how to navigate it, and what sorts of analyses might be pursued.

In addition, since this dataset is a pioneering project in some respects, we see Figure 1 as a model that other researchers interested in publishing qualitative or mixed-methods datasets can use in preparing their own material for publication.

2. Figures 2 to 7 are all descriptions of population baseline characteristics. In other words, from line 396 to line 559, such a long content is only a general description of population baselines in scientific research papers. In other words, it is possible that a table can present all the content more concisely. Of course, as a detailed introduction to interdisciplinary writing or projects, this arrangement may not be unreasonable.

RESPONSE: Thank you for this observation. Given the interdisciplinary nature of our research project and our hopes that the dataset will be engaged by interdisciplinary scholars, we see this descriptive section as a valuable element of the manuscript, as suggested.

3. Traditional research is either prospective or retrospective, but how does this study ensure the authenticity of the records? What does it mean that if someone records documents or videos that do not match reality, how is the project identified? After all, people may have different appearances in front of the camera and in real life.

RESPONSE: Thank you for these questions, which point to a significant difference between conceptions of data in different scientific traditions. In many health fields that study health behaviors or experiences, for instance, researchers recognize from the outset that quantitative data collected will be self-reported, and this factor is taken into account in analyzing data. In keeping with this established tradition of scholarly research, we designed PJP-1 with the recognition that all data collected would be self-reported, with all of the strengths and limitations that this approach entails.

We added a sentence more explicitly clarifying this distinction, and noting the self-reported character of the data and the need to take this into account for (re-)analysis (l. 210–212).

4. As stated in the paper, the project has already begun to be applied in areas such as population mental health, protection of women's and children's rights, vaccine injection, reproductive decision-making, and even immigration and family impact. How are these data quantitatively applied? For example, how is video information converted into data that can be used for textual research?

RESPONSE: Thank you for this question. The study did not collect video data -- only text, image, and audio data. Audio data can be transcribed and analyzed in the same manner as textual data. Images can be catalogued thematically and analyzed in thematic groupings. In short, all three types of data can readily be analyzed using standard qualitative research methods. All three types of data can also be analyzed using quantitative methods like frequency analyses (e.g., to assess the frequency of a particular word, concept, or theme). Other quantitative uses of the data include 1) descriptive statistics of the overall dataset or subsets prepared for specific analyses, and 2) assessment of participant responses to quantitative questions asked either on one occasion or periodically, either independently or (more likely) in conjunction with qualitative data as part of a mixed methods analysis.

5. As an open bidirectional database, how is data security guaranteed? Does it mean that the original creator can change the original record?

RESPONSE: Thank you for the opportunity to clarify these important points. In fact, the PJP-1 database is not open at this point. It is not entirely clear to us what Reviewer #1 means by “bidirectionality.”

To clarify: During the period the study was open to enrollment (May 2020-May 2022), participants were invited to make contributions on a weekly basis via Qualtrics. Data collection ended in May 2022, and the database is now closed. During the study period, participants could, in principle, request that their contributions be removed from our database. Once data collection was complete, contributions could not be changed nor could they be removed by participants.

In terms of data security, all contributions were immediately stored in two locations during the study period. These include 1) the secure research archive we were constructing via Qualtrics, and 2) a mirrored online data archive with two points of interface. One point of interface (“MyJournal”) allowed participants to use a secure log-in to view and/or download their journal entries (and no one else’s). The MyJournal interface has been disabled and decommissioned.

The other point of interface, the curated, publicly accessible Featured Entries, only includes journal entries that are shared with the consent of the participants who contributed them. The Featured Entries page remains open to the public (albeit a different server than the original one).

Data security is guaranteed by the fact that researchers must prepare a formal research proposal; obtain IRB approval from their own institutions; commit to the terms of QDR’s user download agreement; and commit to the terms in our PJP Special Download agreement before obtaining access.

We have added text in l. 390-397 to clarify the stability and security of the described archived data.

---

## [Decision Letter · Decision Letter 1]

19 Aug 2025

The Pandemic Journaling Project: A new dataset of first-person accounts of the COVID-19 pandemic

PONE-D-25-02185R1

Dear Dr. Karcher,

We’re pleased to inform you that your manuscript has been judged scientifically suitable for publication and will be formally accepted for publication once it meets all outstanding technical requirements.

Kind regards,

Benjamin M. Liu, MBBS, PhD, D(ABMM), MB(ASCP)

Academic Editor

PLOS ONE

Additional Editor Comments (optional):

Reviewers' comments:

Reviewer's Responses to Questions

**Comments to the Author**

1. If the authors have adequately addressed your comments raised in a previous round of review and you feel that this manuscript is now acceptable for publication, you may indicate that here to bypass the “Comments to the Author” section, enter your conflict of interest statement in the “Confidential to Editor” section, and submit your "Accept" recommendation.

Reviewer #1: All comments have been addressed

2. Is the manuscript technically sound, and do the data support the conclusions?

Reviewer #1: Yes

3. Has the statistical analysis been performed appropriately and rigorously? 

Reviewer #1: Yes

4. Have the authors made all data underlying the findings in their manuscript fully available?

Reviewer #1: Yes

5. Is the manuscript presented in an intelligible fashion and written in standard English?

Reviewer #1: Yes

6. Review Comments to the Author

Reviewer #1: Thanks the author team for their detailed responses to each of the review comments.

As the authors noted, different disciplines and fields may interpret the same data from different perspectives and using different methods to display. However, this is precisely the charm of interdisciplinary research.

In particular, the integration of traditional medical experimental fields such as virology with new directions in news communication fields like new media may enable researchers, and even those affected by the pandemic, to objectively understand the long-term impact of the COVID-19 pandemic on the real world from diverse perspectives. This could also facilitate the development of public health policies to better address future unknown pandemics.

7. PLOS authors have the option to publish the peer review history of their article (what does this mean?). If published, this will include your full peer review and any attached files.

Reviewer #1: No

---

## [Editor Report · Acceptance letter]

PONE-D-25-02185R1

PLOS ONE

Dear Dr. Karcher,

I'm pleased to inform you that your manuscript has been deemed suitable for publication in PLOS ONE. Congratulations! Your manuscript is now being handed over to our production team.

Kind regards,

on behalf of

Dr. Benjamin M. Liu

Academic Editor

PLOS ONE